# Intermedin in Paraventricular Nucleus Attenuates Ang II-Induced Sympathoexcitation through the Inhibition of NADPH Oxidase-Dependent ROS Generation in Obese Rats with Hypertension

**DOI:** 10.3390/ijms20174217

**Published:** 2019-08-28

**Authors:** Ying Kang, Lei Ding, Hangbing Dai, Fangzheng Wang, Hong Zhou, Qing Gao, Xiaoqing Xiong, Feng Zhang, Tianrun Song, Yan Yuan, Guoqing Zhu, Yebo Zhou

**Affiliations:** 1Department of Physiology, Nanjing Medical University, Nanjing 211166, China; 2Department of Pathophysiology, Xuzhou Medical University, Xuzhou 221004, China

**Keywords:** intermedin, paraventricular nucleus, sympathoexcitation, NADPH oxidase, angiotensin II, obesity-related hypertension

## Abstract

Increased reactive oxygen species (ROS) induced by angiotensin II (Ang II) in the paraventricular nucleus (PVN) play a critical role in sympathetic overdrive in hypertension (OH). Intermedin (IMD), a bioactive peptide, has extensive clinically prospects in preventing and treating cardiovascular diseases. The study was designed to test the hypothesis that IMD in the PVN can inhibit the generation of ROS caused by Ang II for attenuating sympathetic nerve activity (SNA) and blood pressure (BP) in rats with obesity-related hypertension (OH). Male Sprague-Dawley rats (160–180 g) were used to induce OH by feeding of a high-fat diet (42% kcal as fat) for 12 weeks. The dynamic changes of sympathetic outflow were evaluated as the alterations of renal sympathetic nerve activity (RSNA) and mean arterial pressure (MAP) responses to certain chemicals. The results showed that the protein expressions of Ang II type 1 receptor (AT1R), calcitonin receptor-like receptor (CRLR) and receptor activity-modifying protein 2 (RAMP2) and RAMP3 were markedly increased, but IMD was much lower in OH rats when compared to control rats. IMD itself microinjection into PVN not only lowered SNA, NADPH oxidase activity and ROS level, but also decreased Ang II-caused sympathetic overdrive, and increased NADPH oxidase activity, ROS levels and mitogen-activated protein kinase/extracellular signal regulated kinase (MAPK/ERK) activation in OH rats. However, those effects were mostly blocked by the adrenomedullin (AM) receptor antagonist AM22-52 pretreatment. The enhancement of SNA caused by Ang II can be significantly attenuated by the pretreatment of AT1R antagonist lorsatan, superoxide scavenger Tempol and NADPH oxidase inhibitor apocynin (Apo) in OH rats. ERK activation inhibitor U0126 in the PVN reversed Ang II-induced enhancement of SNA, and Apo and IMD pretreatment in the PVN decreased Ang II-induced ERK activation. Chronic IMD administration in the PVN resulted in significant reductions in basal SNA and BP in OH rats. Moreover, IMD lowered NADPH oxidase activity and ROS level in the PVN; reduced the protein expressions of AT1R and NADPH oxidase subunits NOX2 and NOX4, and ERK activation in the PVN; and decreased Ang II levels-inducing sympathetic overactivation. These results indicated that IMD via AM receptors in the PVN attenuates SNA and hypertension, and decreases Ang II-induced enhancement of SNA through the inhibition of NADPH oxidase activity and ERK activation.

## 1. Introduction

The highly interrelated mechanisms, such as the sympathetic nervous system, renin-angiotensin aldosterone system and oxidative stress play an important role in the development of cardiovascular diseases in obesity, which in turn end in organ damage [1]. Hypertension is closely related to the prevalence, pathophysiology and morbidity of obesity [2]. Sympathetic overactivation has been shown to promote the elevation of blood pressure (BP) that involves the development of obesity-related hypertension (OH) [3].

The paraventricular nucleus (PVN) of the hypothalamus is an important integrative site for regulating sympathetic outflow in the brain, which affects the sympathetic nerve activity (SNA) and BP [4,5]. The neurons within the PVN can be activated by endogenous angiotensin II (Ang II), reactive oxygen species (ROS), glutamate, and so on, which promote sympathetic overdrive and raises BP in rats with diabetes, chronic heart failure or hypertension [6,7,8]. Oxidative stress, such as the increase of NADPH oxidase-dependent ROS resource observed in the PVN contributes to sympathoexcitation and elevated BP [9]. Ang II via type 1 receptor (AT1R) can stimulate the activation of NADPH oxidase which increases ROS production in the PVN for the heightening of sympathetic nervous system activity [10]. Our previous studies have shown that the increased Ang II, NADPH oxidase activity and ROS in the PVN can increase the sympathetic activation in OH rats [11]. Therefore, the inhibition of actions of Ang II, NADPH oxidase activity and ROS generation in the PVN may be a good strategy to attenuate the elevation of SNA and BP in OH. 

Intermedin (IMD), a member of the calcitonin/calcitonin gene-related peptide (CGRP) family, was discovered in 2004 [12]. At present, no unique receptor has yet been identified for IMD, and it shares the receptor system with CGRP and adrenomedullin (AM) consisting of calcitonin receptor-like receptor (CRLR) and receptor activity-modifying protein1/2/3 (RAMP1/2/3). IMD can bind nonselectively to all 3 CRLR/RAMP complexes (the CRLR/RAMP1 for the CGRP receptor, and the CRLR/RAMP2 or CRLR/RAMP3 complex for the AM receptor) [12]. It is widely distributed in peripheral organs and the central nervous system [12]. CRLR and RAMPs have been found in the paravenctricular nucleus (PVN) of hypothalamus and abundant IMD-like immunoreactivity has been found in the PVN, including parvocellular and magnocellular cells [13,14]. Our previous study demonstrated that PVN microinjection of IMD decreased SNA and BP in 2-kidney 1-clip (2K1C)-induced renovascular hypertensive rats and OH rats [15]. Moreover, some studies have shown that IMD exerts anti-oxidant effects, involving the inhibition of NADPH oxidase or Ang II-induced oxidative stress [16,17,18]. 

Therefore, this study was designed to investigate the roles and mechanisms of IMD in the PVN in basal SNA and Ang II-induced sympathetic activation in OH rats.

## 2. Results

### 2.1. Metabolic and Anatomical Data

At the end of the 12th week, there was a significant increase in plasma insulin, cholesterol and triglycerides levels, as well as body weight and white adipose tissue mass in OH rats (Table 1) compared to control rats. Plasma glucose level was increased, but it was not significant when compared with control rats. Those results were consistent with our previous published results [19].

### 2.2. The Plasma NE Level, SBP and Protein Expressions of AT1R, IMD, CRLR and RAMP1/2/3 in the PVN

The level of plasma NE is often used to evaluate the basal SNA. NE level in plasma (Figure 1A); SBP (Figure 1B) and AT1R, CRLR and RAMP2/3 (Figure 1C,E–H) protein expressions in the PVN were higher in OH rats than in control rats, but IMD (Figure 1D) protein expression in the PVN was markedly decreased in OH rats. The PAMP1 protein expression had no significant difference between the two groups.

### 2.3. The Effects of Ang II or IMD in the PVN on Basal SNA, and IMD Pretreatment on Ang II-Induced Changes in the RSNA and MAP 

The basal SNA was also evaluated by the changes of RSNA and MAP. Compared to the control rats, the results showed that Ang II in the PVN significantly enhanced the basal SNA in OH rats (Figure 2B,C). However, basal SNA was significantly lowered by IMD in the PVN of OH rats (Figure 2D,E). Moreover, IMD pretreatment effectively inhibited Ang II-induced enhancement in RSNA and MAP in OH rats (Figure 3A,B) as shown by the representative recordings in OH rats (Figure 3C), and these effects of IMD were significantly attenuated by AM receptor antagonist AM22-52 application, but not CGRP antagonist CGRP8-37 (Figure 4A–D).

### 2.4. The Effects of the Pretreatment with Saline, Losartan, Tempo, or Apo in the PVN on SNA Response to PVN Microinjection of Ang II

Ang II-induced ROS generation via AT1R in the PVN has a key role in the modulation of the sympathetic outflow and cardiovascular function [20,21], which is also associated with elevated NADPH oxidase activity, a primary source of ROS [20,21]. However, the importance of Ang II in the PVN in SNA is not completely understood in obesity. In this study, microinjection of Ang II (0.3 nM) into the PVN still significantly potentiated basal SNA in OH rats in the PVN. Pretreatment of Losartan, Tempo or APO in the PVN markedly inhibited the enhanced basal SNA response to PVN microinjection of Ang II in OH rats (Figure 4E,F).

### 2.5. The Effects of IMD in the PVN on NADPH Oxidase Activity and ROS Level, and IMD Pretreatment on Ang II-induced Changes in the NADPH Activity and ROS Level

IMD microinjected into PVN significantly reduced the NADPH oxidase activity and ROS level in OH rats when compared with the control rats. Moreover, Ang II in the PVN induced obvious increases in the NADPH oxidase activity and ROS level which were effectively inhibited by IMD pretreatment in the PVN in the OH rats (Figure 5A,B). ROS content measured by DHE staining in the PVN also showed that OH rats had stronger fluorescence intensity labeled with DHE compared with control rats, and a marked increase after Ang II microinjection was found in OH rats (Figure 5C). IMD pretreatment significantly decreased DHE staining in OH rats. The above roles of IMD were also effectively inhibited by AM22-52 (Figure 5A–C).

### 2.6. The Efects of ERK Inhibitor on a Ang II-Induced Increase in Basal SNA and ERK Activation, and the Effect of IMD Pretreatment on Ang II-Induced ERK Activation

Microinjection of the ERK inhibitor U0126 into the PVN significantly decreased the RSNA and MAP in OH rats, and U0126 pretreatment markedly inhibited Ang II-induced sympathetic excitation (Figure 6A,B) and ERK activation (Figure 6C). Moreover, NADPH oxidase inhibitor Apo or IMD pretreatment also effectively reduced Ang II-induced ERK activation (Figure 6C,D), and the IMD′s effect was prevented by the AM receptor antagonist AM22-52 in OH rats (Figure 6E).

### 2.7. The Effects of Chronic IMD Treatment on Sympathetic Nerve Activity, Systolic Blood Pressure, NADPH Activity and ROS Levels in the PVN

The rats implanted by pumps were continued to be fed normal diet or HFD for 4 weeks. The results showed that IMD significantly decreased the plasma NE level, SBP and HR (heart rate, Figure 7A–C) and reduced the NADPH oxidase activity and ROS level in the PVN of OH rats compared to the control rats (Figure 7D,E). The DHE staining also indicated that chronic IMD treatment can obviously reduce the ROS level in the PVN in OH rats (Figure 8A). Chronic IMD treatment in the PVN lowered the HR in OH rats, but acute application of IMD, ADM22-52 or CGRP8-37 into the PVN had no significant effects on HR (data not shown). 

The rats implanted by pumps were continued to be fed normal diet or HFD for 4 weeks. The results showed that IMD significantly decreased the plasma NE level and SBP (Figure 7A,B), and reduced the NADPH oxidase activity and ROS level in the PVN of OH rats compared to the control rats (Figure 7C,D). The DHE staining also indicated that chronic IMD treatment can obviously reduce the ROS level in the PVN in OH rats (Figure 8A).

### 2.8. The Effects of Chronic IMD Treatment on Ang II-Induced Sympathoexcitation, ERK Activation and AT1R, NOX2 and NOX4 Protein Expressions in the PVN

In order to determine whether chronic IMD treatment in the PVN inhibit Ang II-induced sympathoexcitation in OH rats, we microinjected Ang II into the PVN to explore its roles in RSNA and MAP. As shown in Figure 7E,F, the Ang II-induced sympathoexcitatory and pressor effects were markedly decreased in IMD-treated OH rats, and the AT1R, NOX2 and NOX4 protein expressions and ERK activation in the PVN were also significantly reduced when compared to the control rats (Figure 8B–E). The findings suggested that chronic IMD treatment in the PVN attenuates Ang II-induced sympathoexcitation, maybe through the inhibition of AT1R, NOX2 and NOX4 protein expression and ERK activation, but the specific mechanisms of IMD’s inhibitory effects on these protein expressions needs to be further explored. 

## 3. Discussion

In the present study, there were obvious increases in basal SNA, SBP, and protein expressions of AT1R and receptor system CRLR/RAMP2/3 in the PVN of OH rats, but the endogenous protein expression of IMD in the PVN was much lower in OH rats than in control rats. Acute or chronic IMD application into PVN not only effectively inhibited basal SNA and ROS production, but also attenuated Ang II-induced sympathetic overactivation in OH rats. Acute IMD microinjection into the PVN also markedly reduced Ang II-induced increases in NADPH oxidase activity, ROS generation and ERK activation. These beneficial effects of IMD were effectively blocked by pretreatment with AM receptor antagonist AM22-52. Chronic IMD perfusion into PVN can effectively inhibit AT1R, NOX2 and NOX4 protein expressions, and ERK activation.

NADPH oxidase is recognized as an important source of ROS involved in oxidative stress [22,23]. In the central system, ROS derived from NADPH oxidase are closely related to sympathetic overactivation in cardiovascular diseases, such as chronic heart failure and hypertension [24,25]. Our previous study indicated significant increases of NADPH oxidase activity and ROS level in the PVN of OH rats are involved in the sympathetic overexcitation, and IMD in the PVN can lower enhanced SNA in OH rats [11,15]. In this study, we found that IMD in the PVN decreased NADPH oxidase activity and ROS levels, which may be one of the mechanisms for the sympathetic inhibition of IMD in OH. This result also indicated the inhibitory effect of IMD on NADPH oxidase activity may account for IMD’s potent antioxidant action associated with sympathetic inhibition. Although NADPH oxidase is a possible molecular target of the antioxidant action by IMD, the cellular signaling pathway remains to be determined.

It has been well documented that Ang II in the central nervous system plays very important roles in modulating sympathoexcitation, blood pressure, water and salt appetite, and so on [26,27,28]. In the PVN, Ang II regulates sympathetic outflow, which is a part in the pathogenesis of some sympathoexcitation-characterized diseases [29,30]. Ang II via AT1R can stimulate the NADPH oxidase activation, and the NADPH oxidase-derived ROS play a major role in the central autonomic neurons, such as PVN, nucleus tractus solitarius (NTS) and the rostral ventrolateral medulla (RVLM) [10,31,32]. Moreover, we recently reported that microinjection Ang II into the PVN of OH rats can elicit stronger cardiac sympathetic afferent reflex (CSAR, induced by the stimulation of cardiac sympathetic afferents increases the sympathetic outflow and blood pressure) [11], which indicates that Ang II in the PVN plays an important role in the enhancement of SNA of OH. Therefore, the inhibition of excessive sympathetic excitation caused by Ang II may be a good strategy for the prevention of hypertension in obesity. In this study, IMD pretreatment in the PVN markedly attenuated Ang II-induced sympathetic activation, NADPH activity and ROS production, and this effect was effectively blocked by AM22-52, which suggested that IMD goes via AM receptor to inhibit Ang II-induced sympathoexcitation. IMD and AM have similar biological actions [33], and a previous study demonstrated that ADM blocks Ang II-stimulated ROS generation from NADPH oxidase in rat endothelial cells [34]. Moreover, IMD inhibits unilateral ureteral obstruction-induced oxidative stress via the NADPH oxidase mechanism [18], and it exerts an antihypertrophic effect on neonatal cardiomyocytes by reduced levels of superoxide-derived from Ang II-stimulated NADPH oxidase [35]. Vascular smooth muscle cells with small-interfering RNA knockdown of IMD showed significantly increased Ang II-induced ROS [16]. These studies further confirmed our results that IMD in the PVN suppressed the Ang II-stimulated ROS production, perhaps mainly through the inhibition of NADPH oxidase activity as shown by Figure 9. 

MAPK/ERK was more highly expressed in the PVN of SHR by the fourth week [36]. Ang II-mediated activation of MAPK/ERK increased vesicular release of presynaptic glutamate in the brain [37]. Thus, Ang II-induced sympathoexcitation in the PVN, may involve the alteration of excitatory (glutamate and cathecolamines) and inhibitory (GABA) neurotransmitter systems through ROS-ERK signaling pathways [38]. Recent studies demonstrated that ERK signaling in PVN can modulate SNA to affect cardiovascular function [36,39]. For instance, the anti-obesity effect of nesfatin-1 is mediated by hypothalamic ERK-dependent sympathoexcitation in obese animals [39]. Therefore, the inhibition of PVN ERK activation may be involved in the effect of IMD on Ang II -induced sympathoexcitation in OH rats. In periphery-studies, Ang II significantly enhanced ROS levels, which increased the ERK activation, and ROS clearance was shown to inhibit Ang II-induced ERK phosphorylation [40]. Moreover, we recently found that IMD in PVN attenuates TLR4-mediated ERK activation and sympathetic excitation in rats with obesity-related hypertension, which suggests that IMD may inhibit Ang II-induced ERK activation. Indeed, we confirmed it in this study, and this also indicated that IMD can lower SNA through the inhibition of ERK activation in the PVN caused by inflammation or oxidative stress. In this study, we found that ERK activation inhibitor can effectively inhibit Ang II-induced sympathoexcitation and ERK phosphorylation level. PVN IMD application had a similar inhibitory effect on ERK phosphorylation. This result indicates that the inhibition of ROS–ERK signaling pathways may involve the IMD’s effect on Ang II-induced sympathoexcitation.

The receptor system CRLR/RAMP1/2/3 plays a major role in modulating the effects of IMD [41]. In this study, the PVN had higher endogenous protein expressions of CRLR, RAMP2 and RAMP3 in OH rats, but not RAMP1. We also used the CRLR/RAMP1/2/3 antagonist to validate whether the CRLR/RAMP1/2/3 system mediates the inhibitory mechanism of IMD on NADPH oxidase activity. Our present study showed that CRLR/RAMP2/3 antagonist AM22-52 almost completely abolished the beneficial effects of IMD on basal SNA, Ang II-stimulated sympathetic activation, NADPH oxidase activation and ROS generation. These results demonstrate that CRLR/RAMP2/3 system mediates the IMD’s effects on SNA in the PVN. 

In order to confirm the inhibitory role of IMD in the PVN and its possible mechanisms in sympathoexcitation, chronic IMD perfusion was applied into PVN for 4 weeks by a miniosmotic pump in OH rats. We found that IMD prevented the development of basal SNA and SBP. It not only effectively reduced the ROS level and the protein expression of AT1R and NADPH oxidase subunits NOX2 and NOX, but also attenuated the ERK activation and Ang II-induced sympathetic excitation in the PVN. Thus, IMD in the PVN has an inhibitory effect on Ang II-induced sympathoexcitation in OH that may be partially through the inhibition of NADPH oxidase and ERK activation. Excessive sympathetic activity contributes to hypertension and related organ damage in obese persons with hypertension. The present study found that IMD in the PVN attenuated Ang II-induced sympathoexcitation in HFD-induced obese rats with hypertension, which were mediated by the AM receptors in the PVN. The results indicate that IMD can resist Ang II in the PVN plays sympathoinhibitory and antihypertensive roles in OH, which are mediated by AM receptors.

In conclusion, we propose that IMD in the PVN diminishes basal SNA and attenuates Ang II-induced sympathoexcitation through inhibiting the production of NADPH oxidase-derived ROS and ERK activation. IMD might be an effective therapeutic target for OH treatment in clinical practice. Although our present study provided significant results, there are still some limitations. The primary one is that we did not investigate the effect of IMD gene knockdown on PVN ROS production and SNA. Previous study had demonstrated that IMD and CRLR/RAMPs expressions exist in the hypothalamus, especially abundantly in the PVN. Thus, conditional depletion of IMD from PVN cells may provide more reliable in vivo evidence for a protective role of IMD against OH, and requires future investigation.

## 4. Materials and Methods

### 4.1. Experimental Design

Rats were fed by a normal diet or HFD for 12 weeks. Systolic blood pressure (SBP) and body weight were measured in the conscious state. Acute experiments were implemented at the end of the 12th week. PVN microinjection sites were identified by Evans blue in rats undergoing electrophysiological experiment.

#### 4.1.1. Experiment 1

The plasma norepinephrine (NE) levels (*n* = 6 or 7 for each group) and IMD, AT1R, calcitonin receptor-like receptor (CRLR) and receptor activity-modifying protein (RAMP)1, RAMP2 and RAMP3 protein expressions in the PVN (*n* = 3 or 4 for each group) were examined in control and OH rats.

#### 4.1.2. Experiment 2

The change of basal SNA was measured by the renal sympathetic nerve activity (RSNA) and mean arterial pressure (MAP) responses to IMD (50 pM), Ang II (0.3 nM), an AM receptor antagonist AM22-52 (1 nM) or a CGRP receptor antagonist CGRP8-37 (0.2 nM) in control or in OH rats. Each rat received PVN bilateral microinjections. The intervals between applications of drugs (IMD, AM22-52 or CGRP8-37) were at least 2 h to allow complete recovery, except for Ang II, for which 30 min was the interval (*n* = 6 for each chemical). 

#### 4.1.3. Experiment 3

The effects of pretreatment consisting of PVN microinjections of saline, IMD, AM22-52, CGRP8-37, the MAPK/ERK inhibitor U0126 (50 µM), Tempol (20 nM), Apo (1 nM) or losartan (50 nM) on the RSNA and MAP responses to PVN microinjection of Ang II were investigated in OH rats. The PVN microinjection of Ang II was carried out 10 min after pretreatment with saline, Tempol, Apo, losartan or U0126, and PVN microinjection of IMD was carried out 10 min after pretreatment with AM22-52 or CGRP8-37. Whereas Ang II application was carried out 45 min after pretreatment with IMD (*n* = 6 for each pretreatment).

#### 4.1.4. Experiment 4

The effects of IMD (50 pM), AM22-52 (1 nM) pretreatment on Ang II-induced NADPH oxidase activity and ROS level in the PVN were investigated (*n* = 6 or 7 for each pretreatment). Moreover, the effect of IMD (50 pM), AM22-52 (1 nM) or Apo (1 nM) pretreatment on Ang II-induced ERK activation was investigated in the PVN of rats (*n* = 3–4 for each pretreatment). 

#### 4.1.5. Experiment 5

We determined the effects of chronic infusion of IMD with an Alzet micro-osmotic pump (model 1004, Durect Corp., Cupertino, CA) in the PVN on the plasma norepinephrine (NE) level, SBP, protein expressions of AT1, NOX2 and NOX4, ERK activation in the PVN; and PVN Ang II-induced sympathoexcitation in OH rats. The pumps were implanted in OH rats at the end of the 12th week, and the perfusion lasted for 4 weeks as the HFD was continued. 

### 4.2. Animals

In this study, animal experiments were performed on male Sprague-Dawley rats weighing 160–180 g. They were randomly divided into two groups: One received a normal diet (12% kcal as fat: 12% fat, 60% carbohydrate, 28% protein. Trophic Animal Feed Hightech Co. Ltd., Nantong, China), and the other received a high fat diet (HFD, 42% kcal as fat: 42% fat, 43% carbohydrate, 15% protein. Trophic Animal Feed Hightech Co. Ltd., Nantong, China) for 12 or 16 weeks. Rats were housed in a temperature-and humidity-controlled room with a 12-h light-dark cycle and allowed access to rat chow and tap water ad libitum. The experimental procedures complied with the Guide for the Care and Use of Laboratory Animals (NIH publication, Eighth edition, 2011) and were approved by the Animal Experimental Ethics Committee of Animal Core Facility of Nanjing Medical University(1601149-6, November 15 2018). The rats consuming the normal diet were served as the control group, and rats with obesity-related hypertension (OH group) possessing higher weight gains and SBP ≥ 140 mmHg were used for this study. 

### 4.3. SBP Measurements 

The tail artery SBP was measured by using the noninvasive computerized tail-cuff system (NIBP, ADInstruments, Bella Vista, New South Wales, Australia) in conscious rats, as described previously [11,19]. In order to reduce stress and prevent fluctuations in SBP, the rats were trained to acclimatize by measuring the daily SBP for at least 10 d before performing the formal experiment.

### 4.4. General Procedures of the Acute Experiment

The acute experiment was carried out at the end of the 12th week or 16th week. The rats were administered an anesthesia of urethane (800 mg/kg) and α-chloralose (40 mg/kg) mixture intraperitoneally. During the experiment, supplemental doses of anesthetic agents were used for maintaining an adequate depth of anesthesia. After the exposure of the trachea and carotid artery by a midline incision in the neck, the trachea was cannulated and connected with a rodent ventilator (Model 683, Harvard Apparatus Inc., Holliston, MA, USA) for mechanical ventilation. The right carotid artery was cannulated with a catheter for the MAP measurement by using a pressure transducer connected with a PowerLab data acquisition system (8/35, ADInstruments, Sydney, Australia).

### 4.5. Recording Sympathetic Nerve Activity (SNA)

The left renal nerve was isolated under anesthesia and the same PowerLab data acquisition system was used to simultaneously record the RSNA. The methods for rectifying recordings and integrated RSNA were analyzed as described previously [11,19].

### 4.6. PVN Microinjection

To locate the PVN, each rat was placed in a stereotaxic frame (Stoelting; Chicago, IL, USA). The stereotaxic coordinates for the PVN were 1.8 mm caudal from bregma, 0.4 mm lateral to the midline, and 7.9 mm below the skull surface according to the rat atlas (Paxinos and Watson). Each side of PVN microinjection was completed in one minute and the volume was 50 nL. At the end of the experiment, the same volume of 2% Evans Blue was injected into each microinjection site for histological identification of the PVN with a microscope. Rats with microinjection sites outside of the PVN were excluded from the data analysis. In this experiment, the effects of IMD and the AM receptor antagonist AM22-52 on the baseline RSNA and MAP peaked at about 40–45 min and lasted 90 min or so, but the CGRP receptor antagonist CGRP8-37 had no obvious effects on SNA within 90 min after microinjection. Only when the baseline got back to the normal level and stable for a while (10 min), the next experiment would be carried out, so we selected the intervals between applications of the three drugs as two hours. A representative photo of microinjection sites in the PVN evaluated by Evans blue was shown in Figure 2A. 

### 4.7. PVN Tissue Microdissection 

Rat brains were sectioned serially in 300 µm increments from the bregma; the PVN tissues were isolated by the use of a punch-out technique with a cryostat, as previously described [11,19]. PVN tissue was stored at −80 °C until analyzed for western blotting or used to determine the NADPH oxidase activity and ROS level.

### 4.8. In Situ Detection of Superoxide Anions

The oxidative fluorescence dye dihydroethidium (DHE) was used to evaluate ROS levels in the PVN. Brain coronal sections (30 µm) were thawed and rehydrated using phosphate-buffered saline and incubated for 10 min with dihydroethidium (1 µM) at 37 °C in the dark, as previously described [11,19]. The oxidative fluorescence intensity was visualized by using confocal microscopy (Zeiss LSM 510, Carl Zeiss, Jena, Thuringia, Germany) at a 585 nm wave length.

### 4.9. Measurement of NADPH Oxidase Activity and ROS Level

The enhanced lucigenin chemiluminescence method was used to determine the activity of PVN NADPH oxidase. ROS level in the PVN was determined by using the lucigenin-derived chemiluminescence method. The methods and procedures were as described previously [11,19].

### 4.10. Western Blotting for the Measurement of AT1R, IMD, CRLR, RAMP1/2/3, NOX2 and NOX4 Protein Expression, and Total and Phosphorylated ERK Levels

Western blotting was used to measure the levels of protein expression of AT1R (ab124734), IMD (ab198273), RAMP1/2/3 ((ab-156575, ab-198276, ab-197372), NOX2 (ab129068) and NOX4 (ab133303), antibodies from Abcam, Burlingame, CA, USA) and ERK (T-ERK (16443-1-AP) antibody from Proteintech Group, Inc. Rosemont, IL, USA; P-ERK (AP1015) and CRLR (DF10203) antibodies from Affinity Biosciences. Cincinnati, OH, USA) activation in the PVN. The methods were as described previously [11,19]. Protein loading was controlled by probing all blots with GAPDH antibody (antibody from Bioworld Technology, Louis Park, MN, USA) and the protein intensity was normalized to that of GAPDH.

### 4.11. The Application of Chronic Infusion of IMD in PVN

A 4-week Alzet micro-osmotic pump (model 1004, Durect Corp, Cupertino, CA, USA) with an infusion rate of 0.11 µL/h was connected to the infusion cannula by a catheter to deliver IMD or vehicle into the PVN. For bilateral PVN infusion, the two pumps were implanted on one rat at the end of the 12th week after feeding, and the perfusion lasted for 28 days. The effects of chronic PVN infusion of IMD (50 pM) on the SNA, SBP, and so on were determined.

### 4.12. Blood and PVN Samples Preparation

An overdose of sodium pentobarbital was used to anaesthetize each rat by intraperitoneal injection. Plasma samples were obtained by centrifugation of heparinized blood for estimation of circulating norepinephrine (NE) level. The rat brain was removed and frozen with liquid nitrogen. Finally, the plasma and brain were kept at -80 °C until being used. At the PVN level, coronal sections of the brain were obtained by using a cryostat microtome (Leica CM1900-1-1, Wetzlar, Hessen, Germany).

### 4.13. Measurement of Plasma Glucose, Insulin, Cholesterol and Triglyceride Levels

At the end of 12 weeks, all rats fasted overnight for 12 h without a high-fat diet or control diet but had free access to water before experiments. The next morning, about 1.0 mL of blood was collected from tail vein at around 8 o′clock for the measurement of fasting plasma glucose and insulin, cholesterol and triglyceride levels. The methods and procedures were as described previously [11,19].

### 4.14. Measurement of Plasma NE

The manufacturer′s instructions were followed to examine the level of NE in plasma using Elisa method by a kit from R&D systems (Minneapolis, MN, USA). The final solution was read at 450 nm wave length on a microplate reader (ELX800, BioTek, VT, USA).

### 4.15. Chemicals

IMD was from Bachem (Bubendorf, Switzerland), CGRP8-37 was from AnaSpec (Fremont, CA, USA), and AM22-52, Tempo (20 nM), APO (1 nM), Lorsartan (50 nM) and the MAPK/ERK inhibitor U0126 were from Sigma Chemical (St. Louis, MO). The doses of IMD (50 pM), AM22-52 (1 nM), CGRP8-37 (0.2 nM), U0126 (50 µM) and Ang II (0.3 nM) were chosen with reference to our preliminary studies and published researches [11,15,19]. IMD, CGRP8-37, AM22-52, Lorsartan and Ang II were dissolved in 0.9% NaCl. Apo and U0126 were dissolved in 0.9% NaCl containing 1% of dimethyl sulfoxide (DMSO) (Sigma, St. Louis, MO, USA).

### 4.16. Statistical Analysis

Statistical analyses were performed using Prism version 5.0 (GraphPad Software Inc., San Diego, CA, USA). All data are presented as mean ± SEM, and *p* < 0.05 was considered statistically significant. Differences in the mean values between two groups were assessed by two-tailed student’s *t*-tests. One-way or two-way ANOVA was used for data analysis of more than two groups followed by Bonferroni′s post hoc analysis. There were 5 rats (control group: 2 rats; OH: 3 rats) excluded from data analysis because of the microinjection being not within the PVN, and the microinjection sites used for data analysis were within the marginal regions of the PVN undergoing electrophysiological experimentation. To exclude the possibility that the effects of IMD were caused by diffusion to other brain area, the effects of microinjection of IMD (50 pmol) into the anterior hypothalamic area which is adjacent to the PVN were determined in anesthetized rats (*n* = 5 for control group; *n* = 5 for OH group). The coordinates for the anterior hypothalamic area are 1.8 mm caudal to bregma, 1.0 mm lateral to the midline and 9.0 mm ventral to the dorsal surface. We found that microinjection of same dose of IMD into the anterior hypothalamic area, which is adjacent to the PVN, had no significant effects on the changes of the RSNA and MAP in control group or OH group (data not shown).

## Figures and Tables

**Figure 1 ijms-20-04217-f001:**
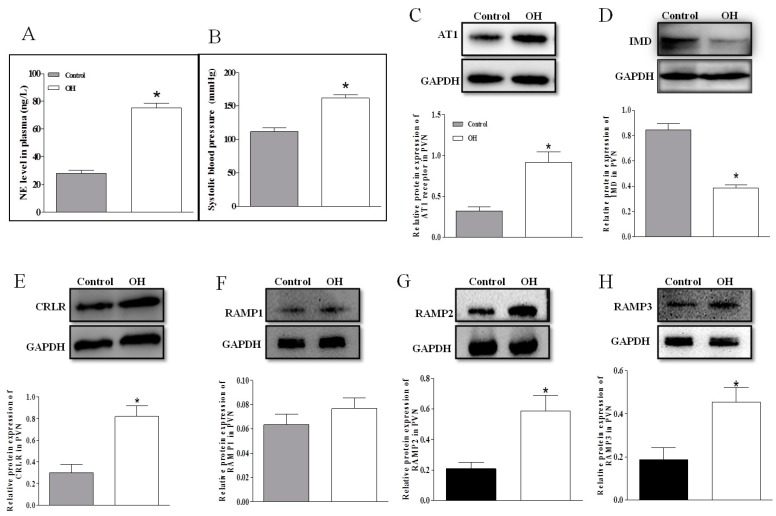
Plasma norepinephrine (NE) level (**A**. *n* = 6 or 7), systolic blood pressure (**B**. *n* = 6–7); and relative values of intermedin (IMD), calcitonin receptor–like receptor (CRLR) and receptor activity–modifying protein (RAMP) 1, RAMP2, and RAMP3 protein expression in the paraventricular nucleus (PVN) in control and obesity-related hypertensive (OH) rats (**C**–**H**. *n* = 3–4). Values are mean ± SEM. * *p* < 0.05 versus control.

**Figure 2 ijms-20-04217-f002:**
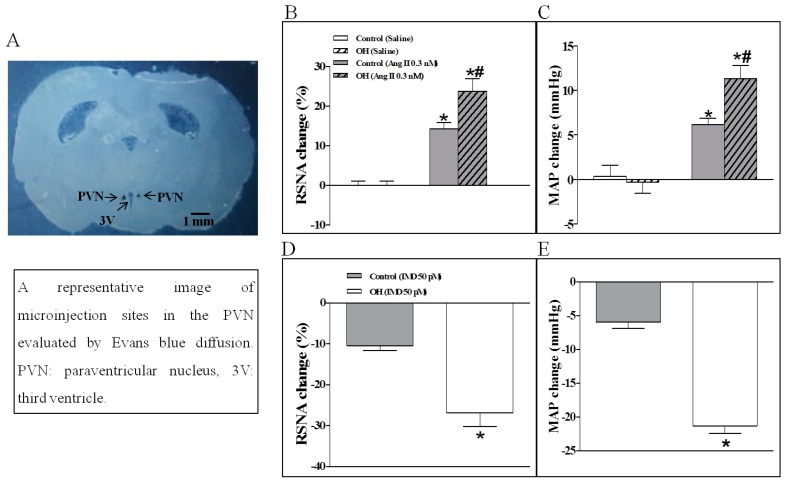
A representative image of microinjection sites in the PVN evaluated by Evans blue diffusion (**A**), and the effects of a PVN microinjection of saline, Angiotensin II (Ang II, 0.3 nM) (**B**,**C**) or intermedin (IMD, 50 pM) (**D**,**E**) on the renal sympathetic nerve activity (RSNA) and mean arterial pressure (MAP) in control and OH rats. Values are mean ± SEM. * *p* < 0.05 versus saline; # *p* < 0.05 versus control (**B**,**C**). * *p* < 0.05 versus control (**D**,**E**). *n* = 6 for each group.

**Figure 3 ijms-20-04217-f003:**
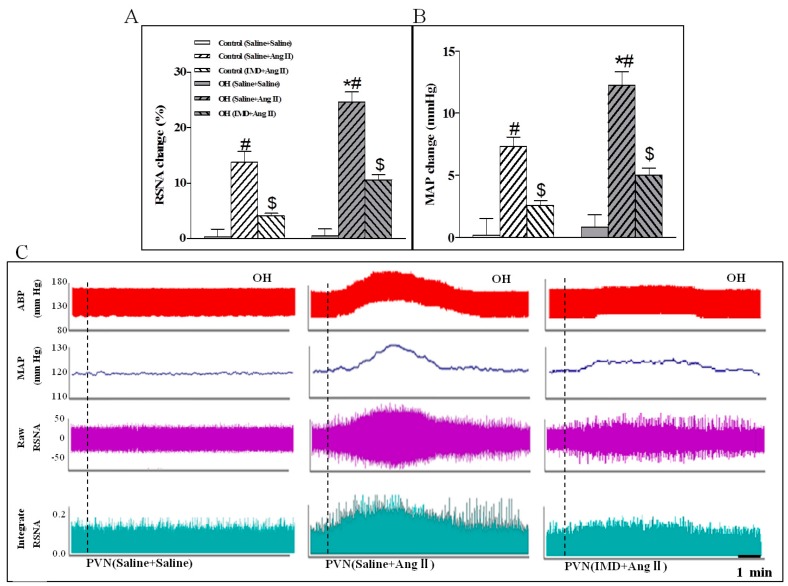
Statistical analysis (**A**,**B**) and representative recordings in OH rats (**C**) showing the effects of saline or IMD (50 pmol) pretreatment in the PVN on RSNA and MAP responses to Ang II (0.3 nM) in control and OH rats. Ang II was administered 45 min after the pretreatment. Values are mean ± SEM. * *p* < 0.05 versus control. # *p* < 0.05 versus saline + saline. $ *p* < 0.05 versus saline + Ang II. *n* = 6 for each group.

**Figure 4 ijms-20-04217-f004:**
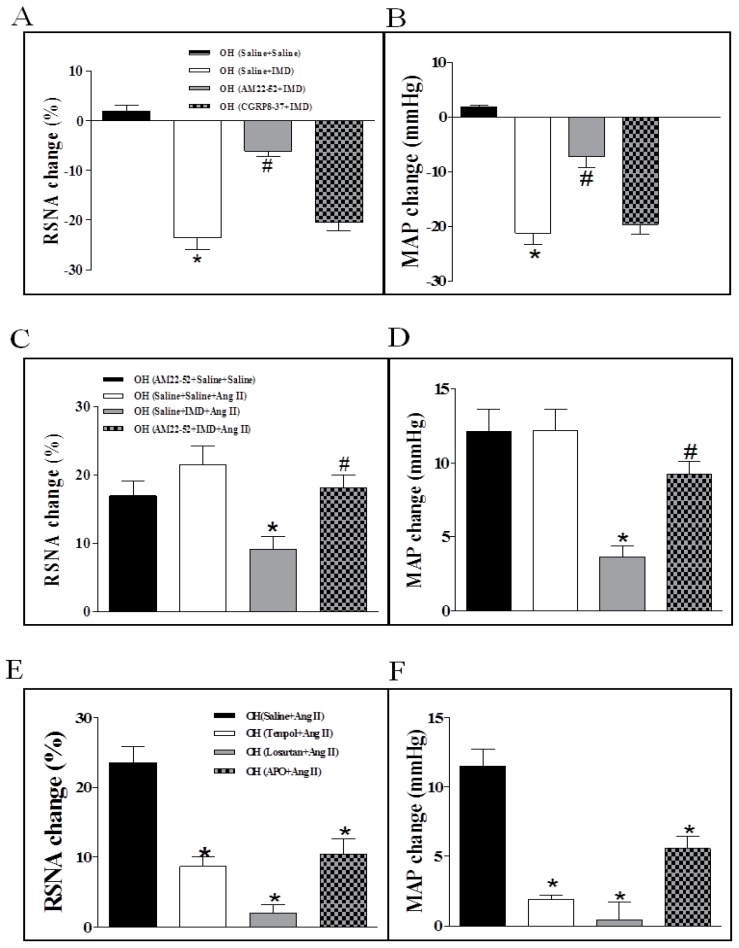
Effects of PVN pretreatment of AM receptor antagonist AM22-52, or CGRP antagonist CGRP8-37 on RSNA and MAP responses to IMD (50 pM) in the PVN of OH rats (**A**,**B**); effects of AM22-52 pretreatment on IMD (50 pM) responses to Ang II (0.3 nM)-induced sympathetic activation in the PVN of OH rats (**C**,**D**); and PVN pretreatment with superoxide scavenger Tempol, AT1R antagonist Lorsatan and NADPH oxidase inhibitor Apo on RSNA and MAP responses to Ang II (0.3 nM) in the PVN of OH rats (**E**,**F**). Values are mean ± SEM. * *p* < 0.05 versus saline + saline; # *p* < 0.05 versus saline + IMD (**A**,**B**). * *p* < 0.05 versus saline + saline + AM22-52 or saline + saline + Ang II; # *p* < 0.05 versus saline + IMD + Ang II (**C**,**D**). * *p* < 0.05 versus saline + Ang II (**E**,**F**). *n* = 6 for each group.

**Figure 5 ijms-20-04217-f005:**
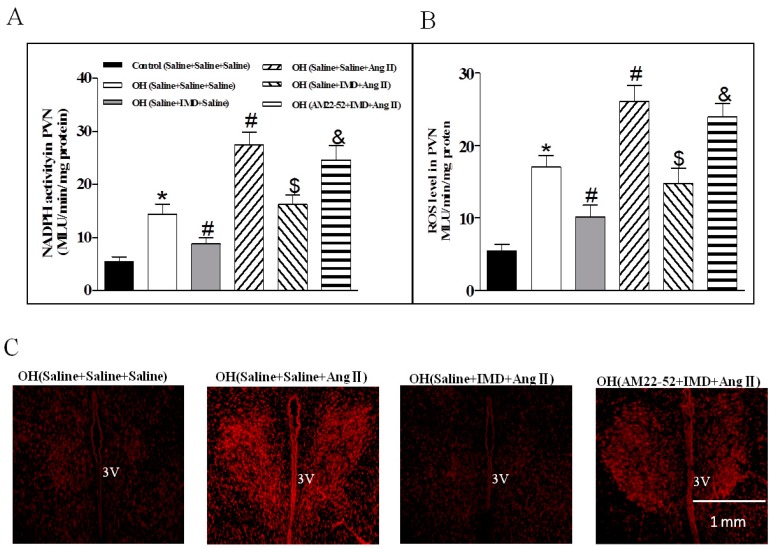
Effects of a PVN microinjection of saline, Ang II (0.3 nM) or IMD (50 pM) on NADPH oxidase activity and ROS levels in the PVN of OH rats; pretreatment of saline or IMD on Ang II-induced NADPH oxidase and ROS levels; and AM receptor antagonist AM22-52 pretreatment on IMD responses to Ang II-induced NADPH oxidase, ROS level and in situ ROS content in the PVN (**A–C**). Values are mean ± SEM. * *p* < 0.05 versus control (saline + saline + saline). # *p* < 0.05 versus OH (saline + saline + saline). $ *p* < 0.05 versus OH (saline + saline + Ang II). *p* < 0.05 versus OH (saline + IMD + Ang II). *n* = 6 or 7 for each group.

**Figure 6 ijms-20-04217-f006:**
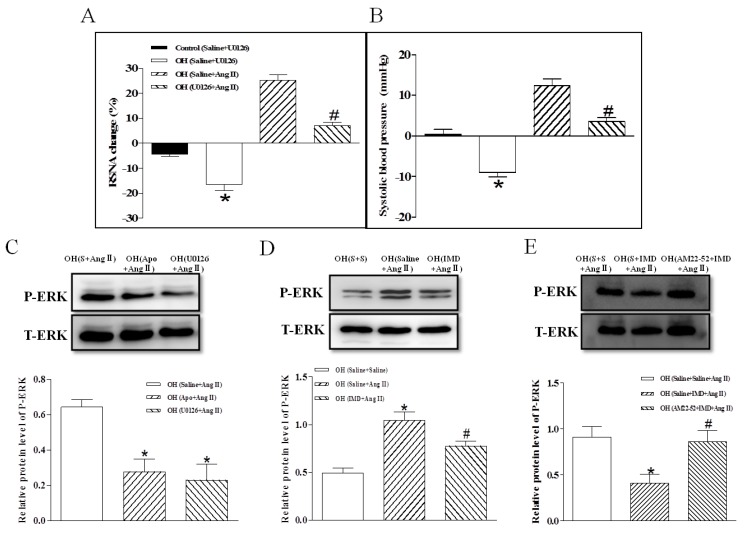
The effects of a PVN microinjection of ERK activation inhibitor U0126 on RSNA and MAP in control and OH rats; PVN U0126 pretreatment on Ang II—induced RSNA and MAP responses in OH rats; Apo, U0126 and IMD pretreatment on Ang II-induced ERK activation in OH rats; and AM receptor antagonist AM22-52 pretreatment on IMD responses to Ang II-induced ERK activation in OH rats. Values are mean ± SEM. * *p* < 0.05 versus control (saline + U0126); # *p* <0.05 versus OH (saline + Ang II) (**A**,**B**. *n* = 6 for each group). * *p* < 0.05 versus OH (saline + Ang II) (**C**. *n* = 3 or 4 for each group). * *p* < 0.05 versus OH (saline + saline + Ang II), # *p* < 0.05 versus OH (saline + IMD + Ang II) (**D**. *n* = 3 or 4 for each group). S: Saline.

**Figure 7 ijms-20-04217-f007:**
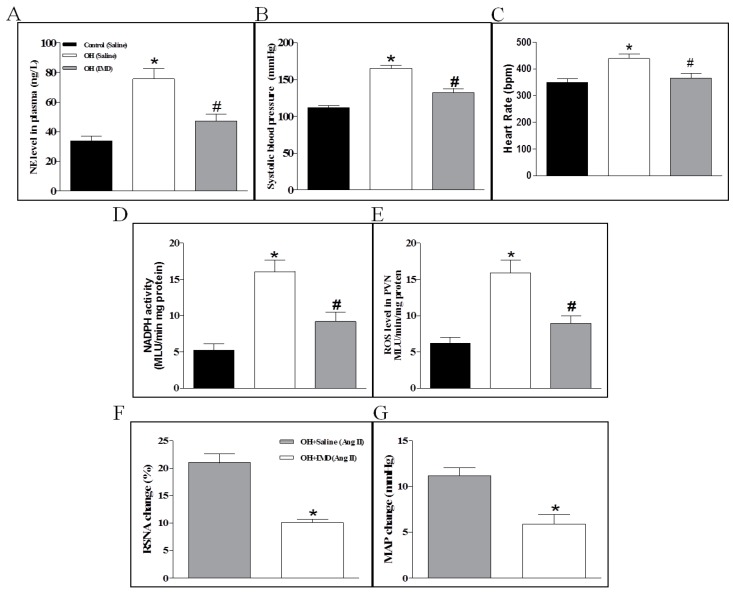
The effects of chronic application of IMD (50 pM, 0.11 µL/hr) in the PVN for 28 days on the plasma NE level (**A**), systolic blood pressure (**B**), PVN NADPH oxidase activity and ROS content in control and OH rats (**C**,**D**); and RSNA and MAP responses to Ang II (0.3 nM) in the PVN of OH rats (**E**,**F**). Values are mean ± SEM. * *p* < 0.05 versus control (saline); # *p* < 0.05 versus OH (saline) (**A**–**D**). * *p* < 0.05 versus OH (saline) (**E**,**F**). *n* = 6 or 7 for each group.

**Figure 8 ijms-20-04217-f008:**
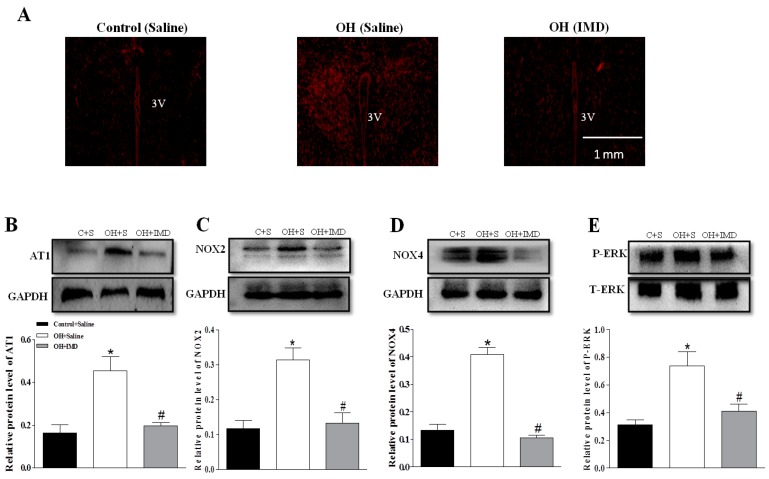
Effects of chronic application of IMD (50 pM, 0.11 µL/hr) in the PVN for 28 days on the protein expressions of AT1R (**B**), NOX2 (**C**) and NOX4 (**D**); ERK activation (**E**); and in situ ROS content (**A**) in the PVN in control and OH rats. Values are mean ± SEM. * *p* < 0.05 versus control (saline); # *p* < 0.05 versus OH (saline) (**B–E**). *n* = 3 or 4 for each group. C: Control, S: Saline.

**Figure 9 ijms-20-04217-f009:**
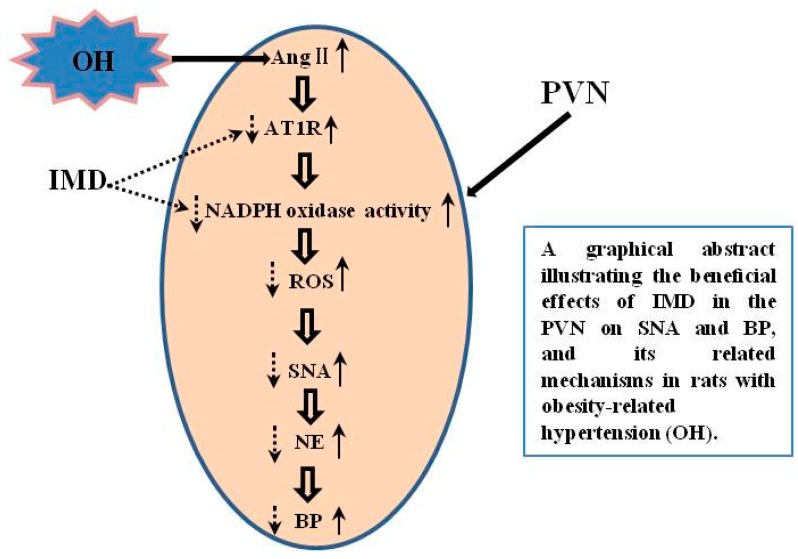
A schematic diagram showing the inhibitory effects of IMD in the PVN on SNA and BP, and its related mechanisms in rats with obesity-related hypertension (OH). IMD effectively prevents the enhanced SNA caused by increased reactive oxygen species (ROS) and improves hypertension via the inhibition of AT1R protein expression and NADPH oxidase activity. The up arrow with dotted line represents increase and the down arrow with solid line represents decrease.

**Table 1 ijms-20-04217-t001:** Metabolic parameters and anatomic data in control and obesity related hypertensive (OH) rats after 12 weeks of a high-fat diet.

Parameters	Control	OH
BW (g)	531 ± 45	653 ± 47 *
Plasma glucose (mg/dL)	139 ± 13	151 ± 16
Plasma insulin (ng/mL)	1.72 ± 0.19	3.07 ± 0.32*
Plasma cholesterol (mg/dL)	47.8 ± 5.2	63.7 ± 5.8*
Plasma triglyceride (mg/dL)	68.2 ± 6.1	95.6 ± 8.7*
Sum of WAT mass (g)	31.9 ± 3.3	61.3 ± 5.9*

OH, rats with obesity-related hypertension induced by a high fat diet; BW, body weight; WAT, white adipose tissue; sum of WAT mass includes inguinal, retroperitoneal, epididymal and mesenteric WAT mass. Values are mean ± SEM; *n* = 10 for each group; * *p* < 0.05 versus control.

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
