# Peer review of "Intermedin in Paraventricular Nucleus Attenuates Ang II-Induced Sympathoexcitation through the Inhibition of NADPH Oxidase-Dependent ROS Generation in Obese Rats with Hypertension"

_ijms, 2019, doi:10.3390/ijms20174217_

Round 1

Reviewer 1 Report

IJMS-567909

The authors aimed to examine whether IMD in the PVN can inhibit the generation of ROS caused by Ang II for attenuating sympathetic nerve activity (SNA) and blood pressure (BP) in rats with obesity-related hypertension (OH) induced by feeding of a high-fat diet (42% kcal as fat) for 12 weeks. The authors showed that IMD via AM receptors in the PVN ameliorated Ang II-induced enhancement of SNA and development of hypertension though the inhibition of NADPH oxidase activity and ERK activation. These results in the present study are interesting for understanding the pathogenesis of hypertension via CNS-mediated mechanism.

Specific comments:
1. The authors of the same research group recently reported that that IMD in the PVN attenuated SNA and hypertension, and also showed that IMD decreased the ERK activation implicated in the LPS-induced enhancement of SNA in the same OH rats induced by feeding of a high-fat diet (42% kcal as fat) for 12 weeks, which was mediated by AM receptors (Sun J, et al. Neurosci Bull. 2019 Feb;35(1):34-46). Thus, the authors should discuss in detail the similarity and novelty over the preceding similar study with overview schema illustrating functional roles of IMD in the PVN via AM receptors in the pathogenesis of LPS- and Ang II-mediated hypertension with obesity.

Author Response

Specific comments:
1. The authors of the same research group recently reported that that IMD in the PVN attenuated SNA and hypertension, and also showed that IMD decreased the ERK activation implicated in the LPS-induced enhancement of SNA in the same OH rats induced by feeding of a high-fat diet (42% kcal as fat) for 12 weeks, which was mediated by AM receptors (Sun J, et al. Neurosci Bull. 2019 Feb;35(1):34-46). Thus, the authors should discuss in detail the similarity and novelty over the preceding similar study with overview schema illustrating functional roles of IMD in the PVN via AM receptors in the pathogenesis of LPS- and Ang II-mediated hypertension with obesity.

Response: Thank you for your suggestions.

Moreover, we recently found that IMD in PVN attenuates TLR4-mediated ERK activation and sympathetic excitation in rats with obesity-related hypertension, which suggests that IMD may inhibit Ang II-induced ERK activation. Indeed, we confirmed it in this study, and this also indicated that IMD can lower SNA through the inhibition of ERK activation in the PVN caused by inflammation or oxidative stress. Excessive sympathetic activity contributes to hypertension and related organ damage in obese persons with hypertension. The present study found that IMD in the PVN attenuated Ang II-induced sympathoexcitation in HFD-induced obese rats with hypertension, which were mediated by the AM receptors in the PVN. The results indicate that IMD can resist Ang II in the PVN plays sympathoinhibitory and antihypertensive roles in OH, which are mediated by AM receptors.

These sentences have been added into the manuscript.

Reviewer 2 Report

The manuscript “Intermedin in paraventricular nucleus attenuates Ang II-induced sympathoexcitation through the inhibition of NADPH oxidase-dependent ROS generation in obese rats with hypertension” by Kang et al. was aimed to elucidate the effects of PVN administration of Intermedin (IMD) on neurogenic hypertension. The authors show that chronic IMD administration via AM receptors in the PVN lowered NADPH oxidase activity and ROS level in the PVN; reduced the protein expressions of AT1R and NADPH oxidase subunits NOX2 and NOX4, and ERK activation resulting in decreased Ang II-induced sympathetic overactivation. Although, the subject matter of this study is of interest the manuscript is scantily written, and there are several issues, primarily related to methodology section and data interpretation, which must be addressed (see the comments below).

Specific recommendations for revision:

Does the intra-paraventricular administration of IMD go to other areas? I assume no (please comment). If yes, would this confound the interpretation of your data because changes in other brain centers may also affect SNA and blood pressure? What was the success rate of the PVN cannula implantation operation?  PBS or secondary antibody controls need to be added in immunofluorescent experiments, and also labeling and legends are confusing for immunohistochemistry data. Figures are not descriptive, and figure legends are poorly written, which make no sense. Western blotting data does not seem to be convincing. Please include uncropped Western blot images with molecular weight ladders visible for peer review and validity of antibodies. Also, clarify whether the bands for GAPDH are matched to the same lanes that are shown for different proteins. Is GAPDH reliable housekeeping for this study? Please comment. No description of Antibodies given. Please provide catalog numbers for each antibody used and some references validating their use. There is controversy in the suitability of commercially available antibody for AT1R. Figure 6– Are levels of total ERK not changing across conditions? And if so, please provide some explanations.  Figures 2, 3, 4, 6, and 7. Please provide the data of HR changes. Please provide representative raw tracings of RSNA, MAP, and HR for data shown in Figures 4A and 4B. These information needs to be added for the accuracy of observations. Experiment 2, Line 301-302. Authors described the intervals between applications of drugs were “at least two hours.” Why were such long intervals chosen, proper justification is recommended. Line 343. Name of anesthetic is missing: “anesthesia of “XXX” (800 mg/kg)”. Which vasculature (or cardiac puncture?) is used to collect blood samples? For measurement of fasting plasma glucose and insulin, please explain fasting condition in detail. How long is fasting? What time of the blood samples were collected? Detailed Information of vehicle (saline or dimethylsulfoxide) should be described. Which chemical is dissolved in saline? Which compounds are dissolved in DMSO? The concentration of DMSO also should be described.

Author Response

Comments and Suggestions for Authors

The manuscript “Intermedin in paraventricular nucleus attenuates Ang II-induced sympathoexcitation through the inhibition of NADPH oxidase-dependent ROS generation in obese rats with hypertension” by Kang et al. was aimed to elucidate the effects of PVN administration of Intermedin (IMD) on neurogenic hypertension. The authors show that chronic IMD administration via AM receptors in the PVN lowered NADPH oxidase activity and ROS level in the PVN; reduced the protein expressions of AT1R and NADPH oxidase subunits NOX2 and NOX4, and ERK activation resulting in decreased Ang II-induced sympathetic overactivation. Although, the subject matter of this study is of interest the manuscript is scantily written, and there are several issues, primarily related to methodology section and data interpretation, which must be addressed (see the comments below).

Specific recommendations for revision:

Does the intra-paraventricular administration of IMD go to other areas? I assume no (please comment). If yes, would this confound the interpretation of your data because changes in other brain centers may also affect SNA and blood pressure? What was the success rate of the PVN cannula implantation operation?  PBS or secondary antibody controls need to be added in immunofluorescent experiments, and also labeling and legends are confusing for immunohistochemistry data. Figures are not descriptive, and figure legends are poorly written, which make no sense. Western blotting data does not seem to be convincing. Please include uncropped Western blot images with molecular weight ladders visible for peer review and validity of antibodies. Also, clarify whether the bands for GAPDH are matched to the same lanes that are shown for different proteins. Is GAPDH reliable housekeeping for this study? Please comment. No description of Antibodies given. Please provide catalog numbers for each antibody used and some references validating their use. There is controversy in the suitability of commercially available antibody for AT1R. Figure 6– Are levels of total ERK not changing across conditions? And if so, please provide some explanations.  Figures 2, 3, 4, 6, and 7. Please provide the data of HR changes. Please provide representative raw tracings of RSNA, MAP, and HR for data shown in Figures 4A and 4B. These information needs to be added for the accuracy of observations. Experiment 2, Line 301-302. Authors described the intervals between applications of drugs were “at least two hours.” Why were such long intervals chosen, proper justification is recommended. Line 343. Name of anesthetic is missing: “anesthesia of “XXX” (800 mg/kg)”. Which vasculature (or â…¡cardiac puncture?) is used to collect blood samples? For measurement of fasting plasma glucose and insulin, please explain fasting condition in detail. How long is fasting? What time of the blood samples were collected? Detailed Information of vehicle (saline or dimethylsulfoxide) should be described. Which chemical is dissolved in saline? Which compounds are dissolved in DMSO? The concentration of DMSO also should be described.

Does the intra-paraventricular administration of IMD go to other areas? I assume no (please comment). If yes, would this confound the interpretation of your data because changes in other brain centers may also affect SNA and blood pressure?

Responses:Thank you very much for your good question. In the “Methods” section, we mentioned that “At the end of the experiment, the same volume of 2% Evans Blue was injected into each microinjection site for histological identification of the PVN with a microscope. Rats with microinjection sites outside of the PVN were excluded from the data analysis.” We are so sorry that the number of rats with microinjection sites outside of the PVN was not written into the manuscript. In fact, there were 5 rats (Control group: 2 rats; OH: 3 rats) were excluded from data analysis because of the microinjection being not within the PVN, and the microinjection sites used for data analysis were within the marginal regions of the PVN undergoing electrophysiological experiment. To exclude the possibility that the effects of IMD were caused by diffusion to other brain area, the effects of microinjection of IMD (50 pmol) into the anterior hypothalamic area which is adjacent to the PVN were determined in anesthetized rats (n= 5 for control group; n=5 for OH group). The coordinates for the anterior hypothalamic area are 1.8 mm caudal to bregma, 1.0 mm lateral to the midline and 9.0 mm ventral to the dorsal surface. We found that microinjection of same dose of IMD into the anterior hypothalamic area, which is adjacent to the PVN, had no significant effects on the changes of the RSNA and MAP in control group or OH group, data not shown.”

What was the success rate of the PVN cannula implantation operation? Responses:PVN cannula traces can be seen with the naked eye when PVN slices were performed for ROS staining or PVN sampling was carried out for western blot experiments, NADPH activity and ROS level measurements. If the cannula trace was not within the PVN of rat, the rat would be excluded. There is only two rats in control groups were excluded because the cannula traces were not within the PVN, but we are not sure the success rate because some rats were performed the electrophysiological experiment, the canula traces in the PVN of these rats maybe blurry because of Angâ…¡injection or Evans blue injection.

PBS or secondary antibody controls need to be added in immunofluorescent experiments, and also labeling and legends are confusing for immunohistochemistry data.

Responses:Thanks for your suggestion. In immunofluorescent experiments, we used the saline to replace the PBS or secondary antibody, the reason is that IMD was dissolved in Saline not PBS or secondary antibody in this experiment. The IMD is an active peptide and it is easily dissolved in Saline. We removed the labeling of PVN and arrows and rewrited the legends in order to avoid confusing.

Figures are not descriptive, and figure legends are poorly written, which make no sense.

Responses:Thank you for your comments! According to your suggestion, the figure legends were rewritten in this manuscript.

Western blotting data does not seem to be convincing. Please include uncropped Western blot images with molecular weight ladders visible for peer review and validity of antibodies. Also, clarify whether the bands for GAPDH are matched to the same lanes that are shown for different proteins.

Responses:Thank you for your suggestions. My students forgot to take a picture of marker when taking photographs for western bands, so most western images in this manuscript have no markers. We are so sorry for the carelessness. According to the reviewer’s suggestion, we know that it is necessary to repeat the Western bloting experiment. At present, we have finished the experiment and selected the representative images most with molecular weight ladders to replace previous images. The most uncropped Western blot images with molecular weight ladders visible were provided and the selected images in this manuscript were marked with red color. The bands for GAPDH were matched to the same lanes that are shown for different proteins.in attachment.

Is GAPDH reliable housekeeping for this study? Please comment.

Responses:Thank you for your question. Recently, we have published two papers which used the same model, and GAPDH protein expression had no significant difference between control group and OH group in the two papers. So in this experiment, we selected GAPDH as housekeeping.

No description of Antibodies given. Please provide catalog numbers for each antibody used and some references validating their use. There is controversy in the suitability of commercially available antibody for AT1R.

Responses:Thank you for your suggestions. We have added the description of Antibodies, namely catalog numbers, into the manuscript. Some information or some references were provided as following:

ADM2(IMD-ab198273):

Description:

Rabbit polyclonal to ADM2/AM2

CRLR(CAT#DF10203):

CALCRL Antibody

CAT: DF10203

Source: Rabbit

Application: WB,ELISA

Reaction: Human,Mouse,Rat

Specificity:

CALCRL (CRLR) Antibody detects endogenous levels of total CALCRL.

RAMP1(ab-156575):

Aubdool AA  et al.A Novel a-Calcitonin Gene-Related Peptide Analogue Protects Against End-Organ Damage in Experimental Hypertension, Cardiac Hypertrophy, and Heart Failure. Circulation136:367-383 (2017). Muschter D  et al.Sensory Neuropeptides and their Receptors Participate in Mechano-Regulation of Murine Macrophages. Int J Mol Sci 20:N/A (2019).

RAMP2(ab-198276):

Description:

Rabbit polyclonal to RAMP2

Application:

WB

RAMP2

RAMP3(ab-197372) :

Fu W  et al.Role of microglial amylin receptors in mediating beta amyloid (Aß)-induced inflammation. J Neuroinflammation 14:199 (2017).

P-ERK(AP1015):

Reference Citations:

1). Liu Y et al. LncRNA-TCONS_00034812 in cell proliferation and apoptosis of pulmonary artery smooth muscle cells and its mechanism. J Cell Physiol 2018 Jun;233(6):4801-4814 (PubMed: 29150946) [IF=4.522]

2). Fan H et al. The in vitro and in vivo anti-inflammatory effect of osthole, the major natural coumarin from Cnidium monnieri (L.) Cuss, via the blocking of the activation of the NF-κB and MAPK/p38 pathways. Phytomedicine 2019 Feb 18;58:152864 (PubMed: 30878874) [IF=4.180]

3). Lu G et al. Inhibition of the cyclophilin A-CD147 interaction attenuates right ventricular injury and dysfunction after acute pulmonary embolism in rats. J Biol Chem 2018 Aug 3;293(31):12199-12208 (PubMed: 29914983) [IF=4.106]

4). Lu G et al. Inhibition of the cyclophilin A-CD147 interaction attenuates right ventricular injury and dysfunction after acute pulmonary embolism in rats. J Biol Chem 2018 Aug 3;293(31):12199-12208 (PubMed: 29914983) [IF=4.106]

5). Xie J et al. 5-aminolevulinic acid photodynamic therapy reduces HPV viral load via autophagy and apoptosis by modulating Ras/Raf/MEK/ERK and PI3K/AKT pathways in HeLa cells. J Photochem Photobiol B 2019 Mar 21;194:46-55 (PubMed: 30925276)[IF=4.067]

6). Yang F et al. Ligustilide, a major bioactive component of Angelica sinensis, promotes bone formation via the GPR30/EGFR pathway. Sci Rep 2019 May 6;9(1):6991(PubMed: 31061445) [IF=4.011]

7). Zhao H et al. CCL17-CCR4 axis promotes metastasis via ERK/MMP13 pathway in bladder cancer. J Cell Biochem 2018 Sep 19 (PubMed: 30230587)

T-ERK(16443-1-AP):

Showing 1- [5] of [109] Publications

Author:

Wang Cai-Ping CP

Pubmed ID:

24099731

Journal:

Neurochem Int

Title:

Isoquercetin protects cortical neurons from oxygen-glucose deprivation-reperfusion induced injury via suppression of TLR4-NF-кB signal pathway.

Application:

WB

Species:

rat

Author:

Su Tzu-Rong TR

Pubmed ID:

24129178

Journal:

Int J Mol Sci

Title:

Inhibition of Melanogenesis by Gallic Acid: Possible Involvement of the PI3K/Akt, MEK/ERK and Wnt/β-Catenin Signaling Pathways in B16F10 Cells.

Application:

WB

Species:

mouse

Author:

Lijing Zhao

Pubmed ID:

24600487

Journal:

Int J Clin Exp Med

Title:

mTOR inhibitor AZD8055 inhibits proliferation and induces apoptosis in laryngeal carcinoma.

Application:

WB

Species:

human

Author:

Cai-Ping Wang

Pubmed ID:

24687774

Journal:

J Neurosci Res

Title:

Mulberroside a protects against ischemic impairment in primary culture of rat cortical neurons after oxygen-glucose deprivation followed by reperfusion.

Application:

WB

Species:

Author:

Cai-Ping Wang

Pubmed ID:

24986222

Journal:

J Physiol Biochem

Title:

Neuroprotective effect of schizandrin A on oxygen and glucose deprivation/reperfusion-induced cell injury in primary culture of rat cortical neurons.

Application:

WB

AT1R(ab124734):

Bian J  et al.Limited AT1 Receptor Internalization Is a Novel Mechanism Underlying Sustained Vasoconstriction Induced by AT1 Receptor Autoantibody From Preeclampsia. J Am Heart Assoc8:e011179 (2019).PubMed: 30845870 Su H  et al.LncRNA H19 promotes the proliferation of pulmonary artery smooth muscle cells through AT1R via sponging let-7b in monocrotaline-induced pulmonary arterial hypertension. Respir Res19:254 (2018).PubMed: 30547791 Zhang Y  et al.Influence of the interaction between Ac-SDKP and Ang II on the pathogenesis and development of silicotic fibrosis. Mol Med Rep 17:7467-7476 (2018).PubMed: 29620193 Xu J  et al.Vitamin D alleviates lipopolysaccharide-induced acute lung injury via regulation of the renin-angiotensin system. Mol Med Rep 16:7432-7438 (2017).PubMed: 28944831 García Trejo EMÁ  et al.The Beneficial Effects of Allicin in Chronic Kidney Disease Are Comparable to Losartan. Int J Mol Sci 18:N/A (2017).PubMed: 28926934 Singh N  et al.ACE2/Ang-(1-7)/Mas axis stimulates vascular repair-relevant functions of CD34+ cells. Am J Physiol Heart Circ Physiol309:H1697-707 (2015).PubMed: 26386115 Yan J  et al.Long-term effects of maternal diabetes on blood pressure and renal function in rat male offspring. PLoS One 9:e88269 (2014). WB ; Rat .PubMed: 24505458 Yang LX  et al.MicroRNA-155 inhibits angiotensin II-induced vascular smooth muscle cell proliferation. J Renin Angiotensin Aldosterone Syst15:109-116 (2014).

NOX2(ab129068):

Li Z  et al.Protective effects of tetramethylpyrazine analogue Z-11 on cerebral ischemia reperfusion injury. Eur J Pharmacol 844:156-164 (2019).PubMed: 30502344 Chen HH  et al.Antihypertensive Potential of Coenzyme Q10 via Free Radical Scavenging and Enhanced Akt-nNOS Signaling in the Nucleus Tractus Solitarii in Rats. Mol Nutr Food Res 63:e1801042 (2019).PubMed: 30668894 Guan L  et al.5 exposure induces systemic inflammation and oxidative stress in an intracranial atherosclerosis rat model. Environ Toxicol34:530-538 (2019).PubMed: 30672636 Geng Z  et al.FNDC5 attenuates obesity-induced cardiac hypertrophy by inactivating JAK2/STAT3-associated inflammation and oxidative stress. J Transl Med 17:107 (2019).PubMed: 30940158 Li Y  et al.NADPH oxidase 2 inhibitors CPP11G and CPP11H attenuate endothelial cell inflammation & vessel dysfunction and restore mouse hind-limb flow. Redox Biol 22:101143 (2019).PubMed: 30897521 Zhang Y  et al.YiQiFuMai Powder Injection Attenuates Coronary Artery Ligation-Induced Heart Failure Through Improving Mitochondrial Function via Regulating ROS Generation and CaMKII Signaling Pathways. Front Pharmacol 10:381 (2019).PubMed: 31031629 Henríquez-Olguín C  et al.Adaptations to high-intensity interval training in skeletal muscle require NADPH oxidase 2. Redox Biol24:101188 (2019).

NOX4(ab133303):

Oglio R  et al.Participation of NADPH 4 oxidase in thyroid regulation.Mol Cell Endocrinol 480:65-73 (2019).PubMed: 30316800 Witten ML  et al.Early life inhalation exposure to mine tailings dust affects lung development. Toxicol Appl Pharmacol 365:124-132 (2019).PubMed: 30641074 Goh KY  et al.Mitoquinone ameliorates pressure overload-induced cardiac fibrosis and left ventricular dysfunction in mice. Redox Biol21:101100 (2019).PubMed: 30641298 Jiang T  et al.Anagliptin ameliorates high glucose- induced endothelial dysfunction via suppression of NLRP3 inflammasome activation mediated by SIRT1. Mol Immunol 107:54-60 (2019).PubMed: 30660990 Kalainayakan SP  et al.Cyclopamine tartrate, a modulator of hedgehog signaling and mitochondrial respiration, effectively arrests lung tumor growth and progression. Sci Rep 9:1405 (2019).PubMed: 30723259 Zhang J  et al.Tumoral NOX4 recruits M2 tumor-associated macrophages via ROS/PI3K signaling-dependent various cytokine production to promote NSCLC growth. Redox Biol 22:101116 (2019).PubMed: 30769285 Ge X  et al.Inhibition of mitochondrial complex I by rotenone protects against acetaminophen-induced liver injury. Am J Transl Res11:188-198 (2019).PubMed: 30787978 Chen P  et al.Palmitic acid-induced autophagy increases reactive oxygen species via the Ca2+/PKCa/NOX4 pathway and impairs endothelial function in human umbilical vein endothelial cells. Exp Ther Med 17:2425-2432 (2019).PubMed: 30906429 Liang X  et al.Cyclic stretch induced oxidative stress by mitochondrial and NADPH oxidase in retinal pigment epithelial cells.BMC Ophthalmol 19:79 (2019).PubMed: 30885167 Kundumani-Sridharan V  et al.Short-duration hyperoxia causes genotoxicity in mouse lungs: protection by volatile anesthetic isoflurane. Am J Physiol Lung Cell Mol Physiol 316:L903-L917 (2019)

About AT1R:

We got the antibody (Anti-Angiotensin II Type 1 Receptor antibody-ab 124734) from Abcam company (cambridge, MA, USA). And it does speak about the antibody specificity on the company’s website (https://www.abcam.cn/angiotensin-ii-type-1-receptor-antibody-epr3873-ab124734-references) as shown: “We are constantly working hard to ensure we provide our customers with best in class antibodies. As a result of this work we are pleased to now offer this antibody in purified format. We are in the process of updating our datasheets. The purified format is designated 'PUR' on our product labels. If you have any questions regarding this update, please contact our Scientific Support team.

This product is a recombinant rabbit monoclonal antibody.

This antibody is a recombinant rabbit monoclonal antibody, and there are 8 published literature which use this antibody. So we think the specificity maybe ok.

Figure 6– Are levels of total ERK not changing across conditions? And if so, please provide some explanations. Responses:Thanks for your question and suggestion. As you mentioned, there was no significant difference in the protein expression of total ERK across conditions. As we know, only phosphorylated ERKl/2 is active, and the activation of ERK is very important for the signal transduction from cell membrane surface receptor to nucleus. The ERK can be activated by other factors such as proinflammatory cytokines and ROS. ERK is also expressed in the PVN and involved in cardiovascular and autonomic regulation. In this study, ERK activation was required for sympathetic activation in OH. More importantly, PVN blockade of p44/42 MAPK activation attenuates hypertension possibly by decreasing ROS and restoring the balance of neurotransmitters.

Yu Y., Wei S. G., Zhang Z. H., Weiss R. M. & Felder R. B. ERK1/2 MAPK Signaling in Hypothalamic Paraventricular Nucleus Contributes to Sympathetic Excitation in Rats with Heart Failure after Myocardial Infarction. American journal of physiology. Heart and circulatory physiology, 2015 (2016). 

Wei S. G., Yu Y., Zhang Z. H., Weiss R. M. & Felder R. B. Angiotensin II-triggered p44/42 mitogen-activated protein kinase mediates sympathetic excitation in heart failure rats. Hypertension 52, 342–350, (2008).

Viedt C. et al.. Differential activation of mitogen-activated protein kinases in smooth muscle cells by angiotensin II: involvement of p22phox and reactive oxygen species. Arteriosclerosis, thrombosis, and vascular biology 20, 940–948 (2000).

Figures 2, 3, 4, 6, and 7. Please provide the data of HR changes. Please provide representative raw tracings of RSNA, MAP, and HR for data shown in Figures 4A and 4B. These information needs to be added for the accuracy of observations.

Response:Thanks for your suggestions. In fact, we have analyzed the HR data, but there was no difference in HR change after IMD, ADM22-52 and CGRP8-37 acute application in the PVN except chronic application of IMD. So we only revised the Figure 7. Please notice the revised version in the manuscript. The sentence “Chronic IMD treatment in the PVN lowered the HR in OH rats, but acute application of IMD, ADM22-52 or CGRP8-37 into the PVN had no significant effects on HR (data not shown).” has been added into the “Result” section of this manuscript.

Experiment 2, Line 301-302. Authors described the intervals between applications of drugs were “at least two hours.” Why were such long intervals chosen, proper justification is recommended.

Responses:Thanks for your question and suggestion. The “at least two hours.” May be not an exact description. In this experiment, the effects of IMD and the AM receptor antagonist AM22-52 on the baseline RSNA and MAP peaked at about 40~45 min and lasted 90 min or so, but the CGRP receptor antagonist CGRP8-37 had no obvious effects on SNA within 90 min after microinjection. Only the baseline got back to normal level and stable for a while (10 min), the next experiment would be carried out, so we selected the intervals between applications of the three drugs were two hours. These sentences have been added into this manuscript.

Line Name of anesthetic is missing: “anesthesia of “XXX” (800 mg/kg)”. Which (or cardiac puncture?)?

Responses:Thank you very much! We have corrected it as following:

The acute experiment was carried out at the end of the 12th week or 16th week. The rats were anesthetized with urethane (800 mg/kg) and α-chloralose (40 mg/kg) intraperitoneally, and vasculature was used to collect blood samples.

For measurement of fasting plasma glucose and insulin, please explain fasting condition in detail. How long is fasting? What time of the blood samples were collected?

Responses:Thanks for your suggestion and questions. The following sentences have been added into the “Methods” section of this manuscript.

At the end of 12 wks, all rats fasted overnight for 12 h without high-fat diet or control diet but had free access to water before experiments. The next morning, about 1.0 mL of blood was collected from tail vein at around 8 o’clock.

Detailed Information of vehicle (saline or dimethylsulfoxide) should be described. Which chemical is dissolved in saline? Which compounds are dissolved in DMSO? The concentration of DMSO also should be described.

Responses:Thank you very much for your questions and suggestions. We have added these sentences into “Methods” section of the manuscript.

IMD, CGRP8-37, AM22-52, Lorsartan and Ang II were dissolved in 0.9% NaCl. Apo and U0126 were dissolved in 0.9% NaCl containing 1% of dimethyl sulfoxide (Me2SO) (Sigma).

Round 2

Reviewer 1 Report

Thank you very much.